# The *Mycobacterium bovis* BCG GroEL1 Contributes to Isoniazid Tolerance in a Dormant-Like State Model

**DOI:** 10.3390/microorganisms11020286

**Published:** 2023-01-21

**Authors:** Sheng Zeng, Dong Yang, Céline Rens, Véronique Fontaine

**Affiliations:** Microbiology, Bioorganic & Macromolecular Chemistry Research Unit, Faculté de Pharmacie, Université libre de Bruxelles (ULB), Boulevard du Triomphe, 1050 Brussels, Belgium

**Keywords:** mycobacteria, dormancy, isoniazid, GroEL1, drug tolerance

## Abstract

Due to the Mycobacterium tuberculosis complex, including *M. tuberculosis* and *M. bovis*, tuberculosis still causes 1.6 million deaths per year. Therefore, efforts to improve tuberculosis treatment are necessary. We previously showed that the GroEL1 protein is involved in antibiotic intrinsic resistance. Indeed, the *M. bovis* BCG *cpn60.1* gene (encoding GroEL1)-disrupted strain (Δ*cpn60.1*) exhibits higher rifampicin and vancomycin susceptibility due to defective cell wall integrity. Here, we show that during hypoxia-triggered growth stasis, in the Wayne dormancy model, the mutant exhibited comparable rifampicin and ethionamide susceptibility but higher isoniazid susceptibility compared to the wild-type strain. Although the Δ*cpn60.1* strain showed compromised induction of the DosR regulon, growth stasis was achieved, but an ATP burst and a higher reactive oxygen species (ROS) production were observed in the isoniazid-treated Δ*cpn60.1* strain. GroEL1 could contribute to INH tolerance by reducing ROS.

## 1. Introduction

Tuberculosis (TB), which is primarily caused by *Mycobacterium tuberculosis* (Mtb), caused an estimated 1.6 million deaths and 10.6 million new cases of TB in 2021 [1]. Together with the burden of drug-resistant TB, this poses a serious global public health threat [1]. Furthermore, approximately one-third of the world population has a latent infection with Mtb. The latent infection is restrained by the host immune response in the lung in solid structures called granulomas. When the host immune system becomes less efficient, patients develop active TB. Inside granulomas, eventually intracellularly in macrophages, Mtb is believed to be present as a mixture of replicating, non-replicating, and dormant states [2]. Non-replicating persistent (NRP) Mtb can be established either by nutrient starvation in an aerobic environment or by the gradual depletion of oxygen [3,4]. The dormant state is further characterized by low metabolic activity and a transient inability to grow [5,6]. The NRP and dormant bacteria are more tolerant to anti-TB drugs and, consequently, are also more prone to developing drug resistance [3,4,7].

Effective TB control relies on a prolonged chemotherapy regimen using multiple drugs, including isoniazid (INH), rifampicin (RIF), ethambutol, and pyrazinamide, with INH and RIF being two of the most effective bactericidal anti-TB drugs [1]. The mechanisms of action of INH involve the formation of an INH–NAD adduct catalyzed by mycobacterial catalase-peroxidase KatG and the subsequent inhibition of InhA, the enoyl-acyl carrier protein reductase required for the synthesis of mycolic acids [8]. Apart from inhibiting essential cell wall lipid production, the bactericidal activity of INH has been further linked to mycobacterial electron transport chain (ETC) perturbation [9]. RIF targets the β subunit of the RNA polymerase (RpoB) and, therefore, inhibits transcription [10]. Although INH and RIF possess strong bactericidal activity against actively growing mycobacteria, their effect on slow-growing and growth-arrested bacilli, e.g., dormant cells, is poor largely due to the acquisition of tolerance, compromising their chemotherapeutic efficacy [4]. However, these two drugs are usually recommended as TB preventive treatments for people living with HIV, those in clinical risk groups, or those having had household contacts with bacteriologically confirmed pulmonary TB patients [1].

TB drug resistance is an increasingly severe issue, with the cases of multidrug-resistant (MDR)/rifampicin-resistant (RR)-TB estimated to be 0.45 million in 2021 [1]. Resistance, typically caused by inherited resistance-conferring mutations, describes the ability of bacteria to survive and grow at high concentrations of an antibiotic [5]. Besides resistance, tolerance is another bacterial strategy contributing to survival during drug treatment. Tolerance is a term used to describe the ability of bacteria to survive the lethal action of bactericidal antibiotics independently of resistance-conferring mutations. Tolerance to antibiotics leads to infection persistency and, eventually, to drug resistance. Therefore, it has received much attention recently [5,7]. A non-replicating dormant-like state can be induced in vitro by nutrient starvation and oxygen deprivation (i.e., hypoxia) [3,4,5]. Slow growth or growth arrest is known to be associated with antibiotic tolerance. Indeed, in the Wayne dormancy model, in which Mtb is confronted with the gradual depletion of oxygen, bacilli eventually adapt their metabolism to enter into a nonreplicating state and acquire tolerance to INH and RIF [4]. Metabolic reprogramming and metabolic shutdown, which occur in growth-arrested bacilli, have both been recently identified as important factors of mycobacterial acquisition of tolerance [11]. For instance, under stresses such as hypoxia, mycobacteria have been shown to enhance their production of triacylglycerol, mainly through the activity of Tgs1, a dormancy survival regulator (DosR) regulon-encoded protein [12,13,14]. Another example of the tolerance-contributing metabolic reprogramming is the bypass of NADH-generating steps in the tricarboxylic acid cycle (TCA), such as by activating the glyoxylate shunt [15]. This rerouting is believed to reduce the production of NADH, thereby limiting the function of the respiratory chain and the generation of bactericidal levels of reactive oxygen species (ROS) to contribute to tolerance [11]. Consistently, reversion of tolerance has been observed when the function of the bacteria ETC, such as in Mtb, was enhanced, leading to the promotion of adenosine triphosphate (ATP) production and to an increase in oxygen consumption [16,17]. Molecules targeting mycobacterial metabolic reprogramming, which is essential to acquiring full antibiotic tolerance, remain to be identified.

*Mycobacterium bovis* BCG (BCG) is closely related to the pathogenic Mtb. Under stresses such as hypoxia, BCG, like Mtb, similarly arrests its growth, activates the DosR regulon which comprises nearly 50 genes, and acquires antibiotic tolerance [14,18]. Like Mtb, BCG harbors a complex cell wall with a high waxy lipid content. Among the lipids, phthiocerol dimycocerosates (PDIM) and phenolic glycolipids (PGL) are two structurally related lipids located at the outer membrane of the cell wall [19]. Mycobacterial mutants with disruptions in the PDIM/PGL biosynthesis or transport pathways exhibited significantly impaired virulence [20,21]. Indeed, PDIM not only promotes macrophage entry for mycobacteria but also prevents phagosomal acidification, contributes to permeabilizing the phagosomal membranes of infected macrophages together with the key type VII secretion system ESX-1, and prompts infection dissemination by facilitating macrophage necrosis [22,23]. Importantly, PDIM and PGL contribute to mycobacterial cell wall integrity and high impermeability [24]. We reported that loss of the two lipids is linked to impaired BCG biofilms and increased susceptibility in Mtb and BCG to various antibiotics, including rifampicin and vancomycin [25,26].

Intriguingly, the biosynthesis of PDIM and PGL involves the mycobacterial GroEL1 (also known as Cpn60.1) encoded by the *cpn60.1* gene. Indeed, a *cpn60.1 * knockout BCG mutant (Δ*cpn60.1*) does not harbor PDIM and normal PGL in its cell wall [26,27]. Expression of *cpn60.1* is induced under heat shock, pH change, and oxidative stress [28]. Considering the involvement of GroEL1 in the production of virulence lipids (PDIM and PGL) and in reducing mycobacterial antibiotic susceptibility under replicating conditions [26], we utilized in this study the well-established Wayne dormancy model to assess the role of GroEL1 in antibiotic tolerance during growth stasis triggered by hypoxia.

## 2. Materials and Methods

### 2.1. Strains and Cultures

The BCG GL2 strains (issued from the BCG Pasteur strain), including wild-type (WT*)* BCG, ∆*cpn60.1* BCG (∆*cpn60.1*), and the complemented strain (Compl∆*cpn60.1*), were previously described [29]. The Pasteur strains of WT *M. bovis* BCG, PMM137 (Δ*fadD26*, PDIM-/PGL+ [30]), and PMM50 (Δ*ppsE*, PDIM^−^/PGL^−^ [22]) were kindly provided by Christophe Guilhot. These strains were grown in Dubos medium supplemented with 10% (vol/vol) Dubos medium albumin (i.e., Dubos Tween Albumin, DTA). To launch a fresh *M. bovis* BCG culture, one vial of glycerol stock from −80 °C was thawed at room temperature (RT) and centrifuged at 3000–4000× *g* for 5 min. Subsequently, the resulting supernatant was removed, and the cell pellet was resuspended and maintained at 37 °C in DTA. Kanamycin (25 µg/mL) was added in the ∆*cpn60.1* and PMM50 cultures, and for the Compl ∆*cpn60.1* strain cultures, kanamycin (25 µg/mL) and hygromycin B (50 µg/mL) were used.

### 2.2. Wayne Dormancy Model

The Wayne dormancy model is an in vitro culture model for investigating mycobacterial hypoxic dormancy [4]. In this model, bacterial preculture in DTA medium was inoculated in test tubes containing 8 mL DTA medium and a magnetic stirring bar, yielding an approximate initial inoculum of 2 × 10^5^ colony forming units (CFU)/mL [4,18]. The tube was then tightly capped with a sterile silicon cap to block the exchange of fresh air. The tubes were placed on a CIMAREC iTM magnetic stirrer platform (Thermo Fisher Scientific, Merelbeke, Belgium). During the incubation, the stirring bar placed at the bottom of each tube turned at a constant speed (200 rpm). With bacterial growth, the gradual depletion of oxygen in the tubes occurred, as determined by the gradual fading of the methylene blue (3 µg/mL), an oxygen dye, triggering the bacilli to adapt into hypoxic dormancy [4].

### 2.3. Drug Treatment and Viability Determination

In the Wayne dormancy model, drugs (200 μL), at appropriate concentrations, were delivered at day 10 or day 20 using a 1 mL syringe by penetrating the tightly capped silicon cap. After drug delivery, the tubes were parafilmed again and incubated for 5 days at 37 °C, before viability measurement. In some experiments, 5 mM N-acetylcysteine (NAC) was simultaneously added to the culture. Test tubes were finally opened to determine OD600 values using a spectrophotometer, and viability was assessed by plating 10-fold dilution samples (10 or 20 µL) on 7H11 agar plates. Plates were incubated in a moisture box at 37 °C, and colonies were counted after 10–20 days. Counting ceased when readings remained constant. In aerobic cultures, we treated bacteria with an approximately equal inoculum size to that in the Wayne dormancy model, i.e., OD_600_ of 0.2–0.3, using identical drug concentrations and treatment duration.

### 2.4. RNA Extraction, Reverse Transcription, and Real-Time PCR Analysis

*M. bovis* BCG cultures (40–50 mL), grown in the Wayne dormancy model or aerobically for 5 days, were harvested by centrifugation at 6000× *g* for 15 min The pellets, which were resuspended in 1 mL of freshly prepared mixture of methanol and chloroform (1/3, *v*/*v*), were vigorously vortexed for 1 min before the addition of 5 mL TRIzol RNA isolation reagent (Thermo Fisher Scientific, Merelbeke, Belgium). Micro-glass beads (approximately 1–1.5 mL) were then added, followed by shaking and rigorous vortexing for 5 min to disrupt cells. Then, 250 µL chloroform were added to bring its final volume to 1 mL. The mixtures were shaken vigorously by hand for 15 s, followed by incubation at RT for 3 min and centrifugation at 4696× *g* for 15 min at 4 °C. The upper aqueous phases were carefully harvested into RNase-free 50 mL tubes. Then, 2.5 mL 100% isopropanol were added and mixed by vortex. The mixtures were incubated at RT for 10 min, followed by centrifugation at 12,000× *g* for 10 min at 4 °C. The supernatant was carefully removed, and the RNA pellets were washed with 2.5 mL 75% ethanol in DEPC-treated H_2_O, followed by centrifugation. To remove any residual ethanol, RNA was air dried for 10 min before solubilization in 40 µL nuclease-free H_2_O. To remove contaminating genomic DNA, the raw RNA (<10 μg) was then treated repeatedly at 37 °C with DNase in reaction mixtures containing 2 units Turbo DNAse (Thermo Fisher Scientific) and 40 units RNAse inhibitor Rnasin (Promega, Leiden, The Netherlands) in a Turbo DNAse buffer. After Dnase treatment, RNA was purified using E.Z.N.A.TM MicroElute RNA clean up kit (Omega, Norcross, GA, USA), and RNA concentrations were measured by the Nanodrop 1000 (Thermo Fisher Scientific). Absence of genomic DNA was regularly confirmed by negative PCR reactions.

Purified RNA (15–35 ng) was reverse transcribed into cDNA using iScript™ Select cDNA Synthesis Kit and random primers (Bio-Rad, Temse, Belgium). The cDNA was kept at −20 °C.

To monitor gene expression, real-time PCR was performed using Takyon No ROX SYBR 2 × MasterMix blue dTTP kit (Kaneka Eurogentec, Seraing, Belgium). The primer pairs used for the real-time PCR are listed in Table 1. The expression of two independent reference genes, sigA and 23S rRNA, was used for normalization. In each experiment, amplification and melting curves were checked.

### 2.5. ROS Determination

BCG cultures in the Wayne dormancy model (Wayne day 10) were treated with INH for two days, followed by measuring ROS with the dye 2,7-dichlorofluorescin diacetate (10 µM, Merck Life Science, Hoeilaart, Belgium) for 2 h. The dye-loaded culture was transferred to a black-wall, flat, and clear-bottom 96-well plate to record fluorescence (excitation at 485 nm and emission at 530 nm (20-nm filter) [31]. ROS data were normalized by corresponding viability.

### 2.6. ATP Determination

*M. bovis* BCG cultures in the Wayne dormancy model (Wayne day 10) were treated with INH for two days, followed by assessment of the adenosine triphosphate (ATP) amount using the BacTiter-Glo Microbial cell viability assay kit (Promega). Briefly, 20 µL of bacterial culture were mixed with equal volume of BacTiter-Glo reagent and incubated for 5 min at RT under light protection. The luminescent signals produced by the BacTiter-Glo luciferase are directly proportional to the amount of ATP present in the reaction. The relative light units (RLU) were measured using a Lumat LB 9507 tube luminometer (Berthold technologies, Baden Württemberg, Germany). The luminescence of the untreated culture reactions was also recorded as a control. The RLU data were normalized by dividing the obtained RLU with the corresponding viability factor or with the OD_600_ values.

### 2.7. Statistical Analysis

Figures were prepared using the GraphPad prism 9.0. An unpaired *t*-test was applied for comparisons between two groups. A *p* < 0.05 was considered to be statistically significant. The symbols *, **, ***, and # indicate *p* ˂ 0.05, 0.01, 0.001, and 0.0001, respectively.

## 3. Results and Discussion

### 3.1. GroEL1 Contributes to Isoniazid Tolerance in the Wayne Dormant-like State Model

Since GroEL1 contributes to mycobacterial cell wall impermeability and intrinsic drug resistance [26], we investigated whether the protein could affect drug susceptibility during growth stasis triggered by hypoxia. The WT, Δ*cpn60.1*, and complemented Δ*cpn60.1* BCG were cultured in the Wayne dormancy model [4]. In this culture model, the bacilli ceased aerobic growth after 4–5 days and entered into a nonreplicating state due to a decreased oxygen content [4,18]. Drugs were injected into cultures at Wayne day 10 by penetrating the silicone caps. Five days later, the tubes were opened for viability determination. The hypoxic WT BCG was >130-fold more tolerant to RIF than its replicating aerobic counterpart (Figure 1A and Appendix A, calculated by comparison of their average survival percentages), as reported for the hypoxic *Mtb* [4]. The hypoxic Δ*cpn60.1* strain had a comparable level of survival following RIF treatment (Figure 1A), although it is more susceptible to RIF under aerobic conditions (Appendix A) [26].

When studying INH susceptibility, we observed a particular phenotype. Indeed, the hypoxic Δ*cpn60.1* strain displayed significantly higher (6.4-fold increase) susceptibility to 0.4 µg/mL INH, as calculated by comparing average survival percentages (Figure 1B). Under aerobic conditions, this concentration is a dose four times more important than the minimal inhibitory concentration (MIC). This INH susceptibility increase could only be observed in the condition in which the survival of the INH-treated hypoxic WT strain was at its optimal, i.e., 100%, the point at which the hypoxic WT strain was fully INH tolerant, that is, in the presence of <1 μg/mL INH. Indeed, the difference in INH susceptibility between the WT and Δ*cpn60.1* BCG strains became insignificant when the hypoxic cultures were treated with higher INH concentrations, resulting for the WT strain in lower bacterial survival (Figure 1C). This compromised INH tolerance was also observed in the Δ*cpn60.1* strain when drugs were injected into Wayne cultures at Wayne day 20 (data not shown) and was significantly partially restored in the *cpn60.1* complemented strain (Figure 1B). When studying the susceptibility of the strains to another antibiotic targeting mycolate synthesis, ethionamide (ETH), WT, and Δ*cpn60.1* strains were equally susceptible (Figure 1D). The difference seen in the presence of 0.4 µg/mL INH was, thus, not linked to mycolate synthesis inhibition. This difference was also observed only under hypoxia, as the WT and Δ*cpn60.1* strains were equally susceptible to INH under aerobic replicating conditions (Appendix A).

### 3.2. GroEL1 Is Not Required for Hypoxic Growth Arrest but Contributes to the DosR Regulon Optimal Induction

In order to understand the mechanisms of action of GroEL1 in INH tolerance during hypoxia, we first verified whether all the tested BCG strains grown in the Wayne dormancy model could similarly exhibit a growth arrest. Indeed, if the Δ*cpn60.1* strain would not reach the same growth arrest as the WT strain in the Wayne model, it could explain its compromised INH tolerance in this condition. As shown in Figure 2A, the growth pattern of the Δ*cpn60.1* strain was indistinguishable from its parental WT strain. Both stopped replication at Wayne day 4 (Figure 2A,B), ceasing the early exponential phase and entering into a nonreplicating stage [4]. Notably, the *Δcpn60.1* strain maintained its viability at Wayne day 15 and Wayne day 20 (Figure 2B), suggesting that GroEL1 is dispensable for mycobacterial survival under the dormant-like state induced by hypoxia.

Hence, the growth arrest, which is generally associated with bacterial tolerance [5], is not compromised in the hypoxia-adapted Δ*cpn60.1* strain, ruling it out as a main mechanism for the defective INH tolerance observed for this mutant strain, under hypoxia.

The dormancy establishment in mycobacteria is controlled by two sensor kinases (i.e., DosT and DosS) together with a response regulator DosR, whose activation triggers the expression of nearly 50 regulon genes, including hspX and fdxA [14]. To investigate whether the Δ*cpn60.1* strain could equally activate the DosR regulon under hypoxia, the levels of mRNA of representative regulon genes were assessed by quantitative PCR for BCG Wayne cultures at Day 5. As expected, the induction of the DosR regulon in the WT BCG, relative to its aerobic replicating condition, was significant under hypoxia, as suggested by the 9.3-fold induction of dosR, the 4.5-fold induction of dosS, the 15-fold induction of dosT, the 186-fold induction of hspX, and the 198-fold induction of fdxA (Figure 3). Although the level of mRNA encoding DosR (2.8-fold), DosT (2.3-fold), FdxA (26-fold), and HspX (19-fold) was upregulated in the Δ*cpn60.1* strain compared to its aerobic replicating condition (Figure 3), their induction levels were much lower compared to that of the WT strain (Figure 3). Moreover, the hypoxic induction of dosS, which encodes a redox sensor whose function is reported to be regulated by mycobacterial ETC status [32,33], was marginal (1.2-fold relative to the aerobic condition, Figure 3). This suggests a reduced DosR regulon induction during hypoxia in the absence of GroEL1, although this difference failed to translate into a survival defect (Figure 2B). The expression of *phoP*, which encodes a regulator able to modulate dosR/dosS expression and potentially to participate in the hypoxic response [34], was not induced in both WT and Δ*cpn60.1* strains under the tested condition (Figure 3) and could, thus, serve as a control for the assay. Similar results were obtained when another reference gene, i.e., 23S rRNA, was used to normalize.

In addition, the expression of the sensor kinase DosS, as well as other regulon proteins, was also significantly decreased in the Δ*cpn60.1* biofilm cells [25], suggesting a general role for the protein in the optimal induction of mycobacterial DosR regulon under various stresses (Figure 3). It is important to note that mycobacterial biofilms, similar to hypoxia, contribute to the acquisition of antibiotic tolerance [25,35,36] and that the normal function of the DosR regulon is necessary for acquiring antibiotic tolerance [37]. For instance, under hypoxia, mycobacteria activate the expression of one of the DosR regulon genes, tgs1, to enhance the synthesis of triacylglycerol [12,13,14]. The intracellular accumulation of this lipid is important for both growth arrest and antibiotic tolerance [13,15]. Hence, the compromised INH tolerance observed for the hypoxic Δ*cpn60.1* strain (Figure 1B) could be partially due to the suboptimal induction of the regulon. The DosR regulon regulation by GroEL1 may be an indirect effect. We previously showed that the protein participates in mycobacterial respiratory downregulation and contributes to lower ETC activity under stresses [25]. We reason that the GroEL1-facilitated ETC downregulation may alter the cellular redox balance to a more reduced state (e.g., a higher menaquinol/menaquinone ratio), which was shown to trigger the induction of the DosR regulon [33,38]. Further assays, such as the assessment of the redox state of the hypoxic Δ*cpn60.1* strain, are required to test this hypothesis.

A previous report showed that in *Mtb*, the loss of GroEL1 reduced by twofold bacilli survival under low aeration [39]. It is possible that in our study, the smaller initial inoculum and the gradual loss of oxygen used in our model could have facilitated the sequential adaptation of the Δ*cpn60.1* strain into a dormant-like state (however, it was suboptimal compared to the WT strain, given the compromised DosR regulon induction), allowing the mutant bacteria to better survive.

### 3.3. PDIM Doesn’t Contribute to Isoniazid Tolerance during the Dormant-like State Induced by Hypoxia

Since the Δ*cpn60.1* strain does not harbor the PDIM and the same PGL lipids in the outer layer of its cell wall [26,27], we also assessed whether two BCG mutants issued from the WT 1173P2 Pasteur strains PMM50 and PMM137 could also have a defective INH tolerance in the Wayne dormancy model. The PMM50 strain lacks both PDIM and PGL [22], whereas the PMM137 strain is devoid of only PDIM [30]. Both mutants behaved similarly in the Wayne dormancy model compared to the WT Pasteur strain (Appendix A), indicating that the virulence lipids, PDIM and PGL, are not important for the dormant-like growth arrest induced by hypoxia. Furthermore, the mutants deficient in PDIM and/or PGL developed INH tolerance to the same extent as their WT counterpart in this hypoxic dormancy model (Figure 4). Therefore, the compromised INH tolerance observed for the Δ*cpn60.1* strain does not result from a deficiency of PDIM/PGL in its cell wall.

### 3.4. GroEL1 Contributes to the Metabolic Adaptation Required for INH Tolerance

We previously reported that GroEL1 contributes to mycobacterial ETC downregulation during biofilm growth [25]. ETC downregulation, part of the metabolic reprogramming/downshifting essential for the establishment of tolerance [11], leads to lower ATP generation, resulting further in downregulation of the numerous ATP-consuming events in a cell. Enhancing respiration and ATP generation is linked to the reversion of bacterial tolerance [16,17]. We therefore measured the ATP levels of hypoxic BCG strains following INH treatment. It was observed that the Δ*cpn60.1* strain exhibited a markedly increased ATP amount following INH challenge, a phenotype not observed for the WT strain (Figure 5A). We previously demonstrated that INH-treated aerobic bacilli had an enhanced ETC activity, including an augmented ATP generation, and this was linked to the bactericidal activity of the drug [9]. Therefore, the ATP burst in the INH-treated mutant strain could explain or reflect the compromised tolerance to the drug.

A significantly increased ATP level may reflect a stronger activity of the ETC, the major source of cellular ROS [40]. Furthermore, an increased ROS generation was linked to the reversion of Mtb INH tolerance [17]. The information suggests that the INH-treated hypoxic Δ*cpn60.1* strain could have a higher ROS level, contributing to its compromised INH tolerance. Indeed, although the untreated BCG strains had comparable ROS levels, INH treatment markedly increased the ROS level of the hypoxic Δ*cpn60.1* strain (eightfold increase). This was not observed for the hypoxic WT strain (only a 1.5-fold increase), and the level of ROS in the complemented strain was partially restored (only a 2.8-fold increase) compared to its untreated control, as expected from a complemented mutant strain (Figure 5B). In addition, treatment of hypoxic WT BCG with 4 μg/mL INH, a stronger bactericidal dose, which lead to lower mycobacterial survival and, thus, unsatisfactory tolerance (Figure 1C), also produced a higher amount of ROS (Figure 5C). Interestingly, Sharma and colleagues observed that under hypoxia (but not normoxia), the loss of GroEL1 resulted in decreased production of mycobacterial antioxidant components, such as the alkyl hydroperoxide reductase C and D proteins and the superoxide dismutase [39,41,42,43,44]. It is, therefore, tempting to speculate that the loss of GroEL1 in the Δ*cpn60.1* strain could have compromised mycobacterial antioxidative defense mechanisms, potentially leading to the accumulation of ROS in the INH-treated mutant strain.

To investigate whether the enhanced INH susceptibility of the hypoxic Δ*cpn60.1* strain could be due to higher ROS production, N-acetylcysteine (NAC), an antioxidant [45], was added or not added with INH into the hypoxic Δ*cpn60.1* cultures. The addition of NAC significantly protected the hypoxic Δ*cpn60.1* strain from INH-induced killing (Figure 5D), which restored INH tolerance. We previously reported that the addition of NAC prevents the INH-triggered ATP burst and protects the aerobic bacilli from INH-induced killing [9]. Therefore, the enhanced INH susceptibility of the GroEL1-deficient strain could result from ROS toxicity and/or from the ATP burst, and the latter has recently been linked to INH-induced aerobic killing [9,46]. This notion is also in agreement with previous reports, which demonstrates the importance of ROS generation associated with INH and other antibiotic-induced bactericidal activities [18,47,48,49]. It is worth noting that a loss of HupB, a nucleoid-associated protein, was shown to make Mtb highly susceptible to reduced amounts of RIF and INH. HupB also helps Mtb planktonic aerobic culture to gain tolerance against high antibiotic concentrations [50]. It is, therefore, possible that within the hypoxic growth arrest system of Wayne, GroEL1, also considered to be a nucleoid-associated protein [51], could contribute to protecting DNA from deleterious ROS damage. To verify this hypothesis, it will be interesting in future studies to assess the protective impact of GroEL1 on DNA damage, for example, in a comet assay. However, the mechanisms of action of GroEL1 are probably multiple, as this protein has already been shown to be involved in various metabolic pathways, such as in the PDIM/PGL biosynthesis and in respiratory metabolism adaptation, although most probably through indirect effects involving metabolic enzyme folding [25,26].

## 4. Conclusions

We showed that GroEL1, PDIM, and PGL are not essential for mycobacterial hypoxic growth arrest in the Wayne model. However, in this dormancy model, GroEL1 contributed to the INH tolerance by allowing optimal dormancy regulon induction, reducing the amount of ATP, and limiting ROS levels. This observation pinpoints for the first time the role of GroEL1 in INH tolerance in a dormant-like state. In view of the role of GroEL1 in mycobacterial intrinsic resistance against vancomycin and rifampicin through its essential involvement in PDIM and PGL biosynthesis, the contribution of GroEL1 to INH tolerance during hypoxic growth arrest further supports GroEL1 as an interesting target for novel anti-TB drug development.

## Figures and Tables

**Figure 1 microorganisms-11-00286-f001:**
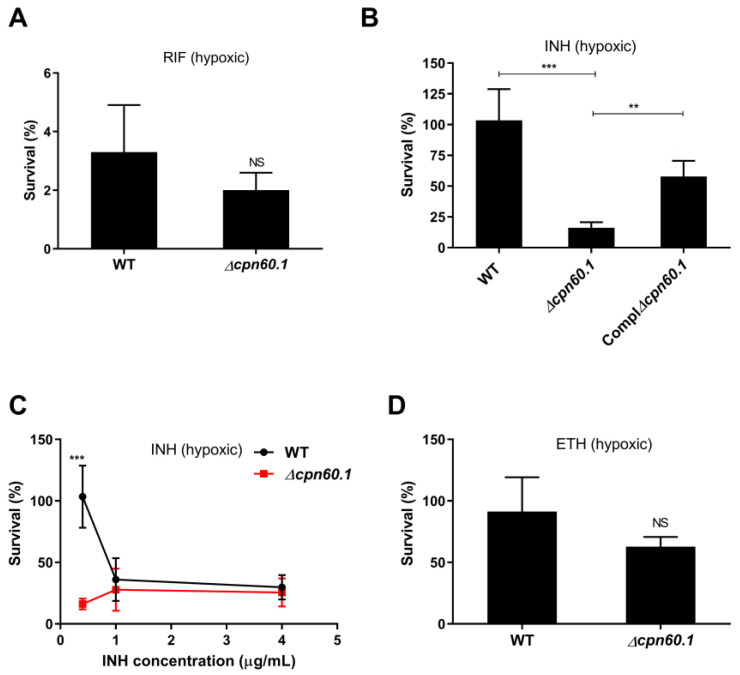
GroEL1 contributes to INH tolerance in the Wayne dormancy model. BCG strains maintained in the Wayne dormancy model were treated at Wayne day 10 with 0.1 µg/mL RIF (**A**); 0.4 µg/mL INH (**B**); 0.4, 1 and 4 µg/mL INH (**C**); and 15 µg/mL ETH (**D**) for 5 days. Viability was determined by plating sample dilutions on 7H11 agar plates and comparing the amount of the obtained CFU after incubation to those obtained with mock treatment. Mean values and standard deviations from three or four experiments are shown. NS—statistically no significant difference; **—*p* ˂ 0.01; ***—*p* ˂ 0.001.

**Figure 2 microorganisms-11-00286-f002:**
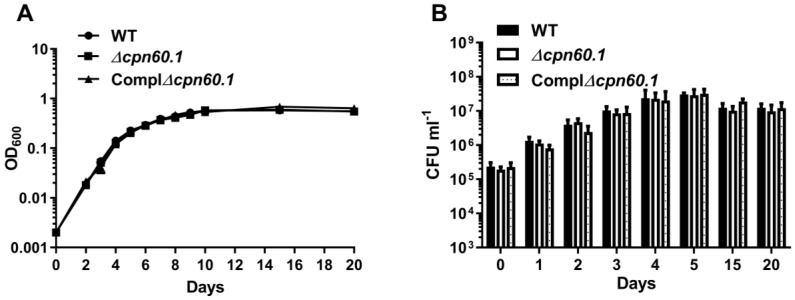
Growth of the WT, Δ*cpn60.1*, and complemented Δ*cpn60.1* BCG strains at various time points in the Wayne dormancy model, as monitored by OD600 (**A**) and bacteria survival (**B**). Bacterial viability was determined by plating sample dilutions on 7H11 agar plates, and CFUs were recorded when readings were consistent. Data were pooled from at least 5 independent experiments.

**Figure 3 microorganisms-11-00286-f003:**
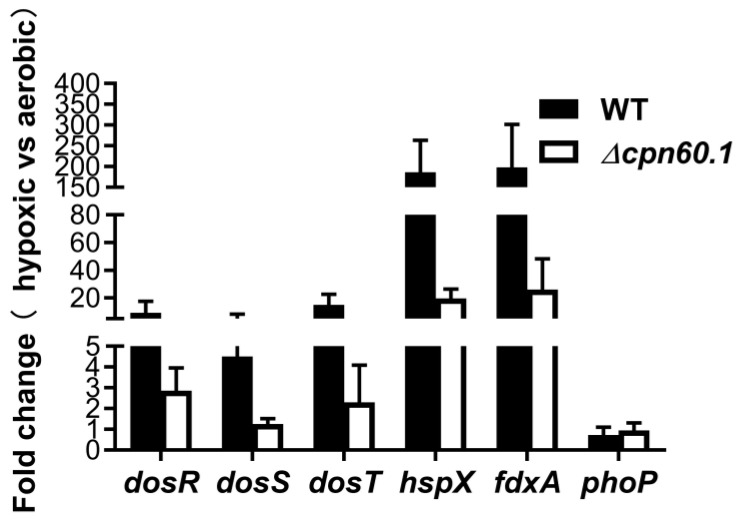
Relative assessment of the induction of various dormancy related genes for the WT and Δ*cpn60.1* strain in the Wayne dormancy model at Wayne day 5. The relative amount of the dosR, dosS, dosT, fdxA, and hspX dormancy induced mRNA and of the phoP mRNA in the WT and in the Δ*cpn60.1* BCG strains in the Wayne dormancy model at day 5 are presented after normalization according to sigA mRNA relative quantification and compared with that of the corresponding mRNA levels obtained from cultures in aerobic condition.

**Figure 4 microorganisms-11-00286-f004:**
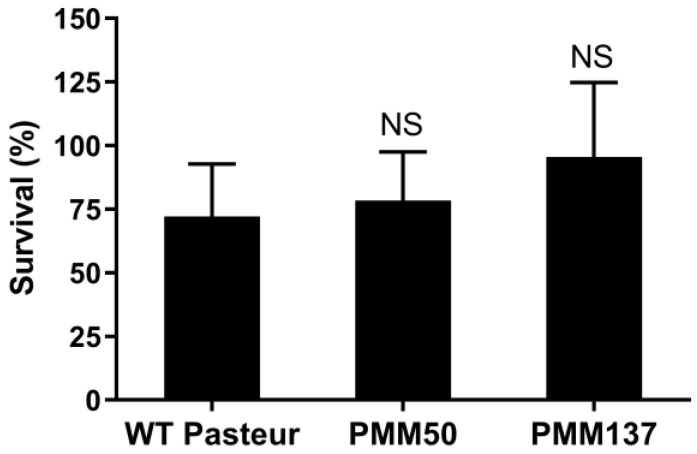
The loss of PDIM in the *M. bovis* BCG cell wall does not interfere with the acquisition of tolerance in the Wayne dormancy model under 0.4 μg/mL INH. Viability was determined by plating sample dilutions on 7H11 agar plates and comparing the amount of obtained CFUs after incubation to those obtained with mock treatment. Mean values and standard deviations from three independent experiments are shown. NS—statistically no significant difference compared to WT Pasteur strain survival level.

**Figure 5 microorganisms-11-00286-f005:**
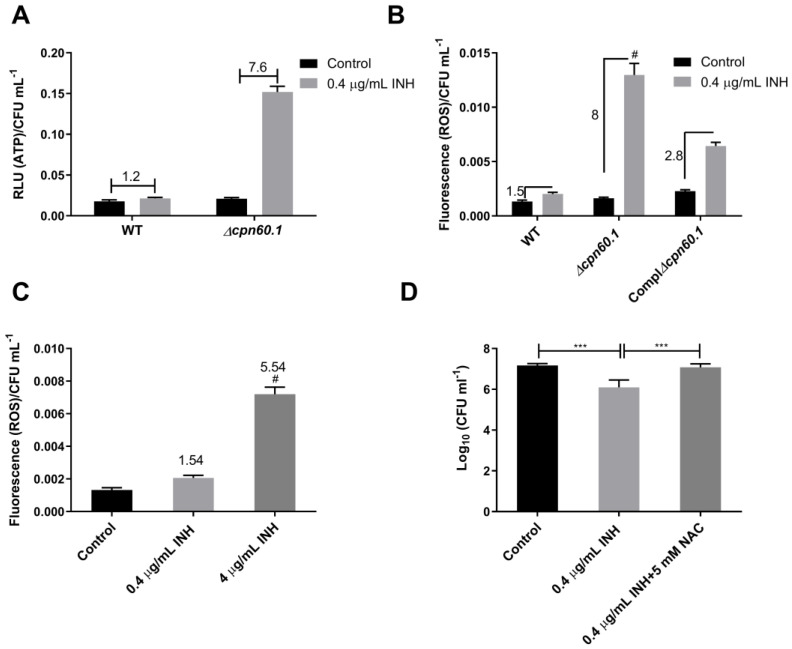
GroEL1 contributes to INH tolerance in the Wayne dormant-like state model by limiting ROS. (**A**) WT and ∆*cpn60.1 M. bovis* BCG cultures grown in the Wayne dormancy model (at day 10) were treated or untreated with 0.4 µg/mL INH for 2 days and harvested for ATP level assessment. ATP levels were assessed using a luciferase activity assay. RLUs were normalized by the number of living bacteria/viability. (**B**) WT, ∆*cpn60.1,* and Compl∆*cpn60.1* BCG cultures at Wayne day 10 were treated with 0.4 µg/mL INH for 2 days, followed by determination of ROS using 2,7-dichlorofluorescin diacetate (10 µM) as a probe. ROS data were normalized by the number of living bacteria/viability. (**C**) WT BCG cultures at Wayne day 10 were treated with 0.4 µg/mL and 4 µg/mL INH for 2 days, followed by determination of ROS using 2, 7-dichlorofluorescin diacetate (10 µM) as a probe. ROS data were normalized by the number of living bacteria/viability. (**D**) Viability of the ∆*cpn60.1* strain in the Wayne dormancy model (at day 10) untreated (control) or treated with 0.4 µg/mL INH for 5 days with or without 5 mM N-acetylcysteine (NAC). Mean values and standard deviations from three or four experiments are shown. *** *p* ˂ 0.001; # *p* ˂ 0.0001.

**Table 1 microorganisms-11-00286-t001:** Primers used in real-time PCR.

Primer Name	Sequence 5′ to 3′
23S forward primer	GAAGAATGAGCCTGCGAGTC
23S reverse primer	GGTCCAGAACACGCCACTAT
sigA forward primer	CTCGGTTCGCGCCTACCTCA
sigA reverse primer	GCGCTCGCTAAGCTCGGTCA
dosR forward primer	CGGTCGCTGCTGGACAATC
dosR reverse primer	TTTCGGCTAGGAACATTCG
dosS forward primer	ACCGGCAGCATCGGGTATTGC
dosS reverse primer	TCGATGAGCAGCCCGATGAC
dosT forward primer	CATCGGTTGGATTTCCGCTG
dosT reverse primer	CCATCTGCCTTCTCGGTCAA
fdxA forward primer	CCTATGTGATCGGTAGTGA
fdxA reverse primer	GGGTTGATGTAGAGCATT
hspX forward primer	CGCACCGAGCAGAAGGAC
hspX reverse primer	CCGCCACCGACACAGTAA
phoP forward primer	TGGGTGGTGACGACTATGTG
phoP reverse primer	GTGGTTCCTTGTTGCCCTTG

## Data Availability

Data suppoting reported results can be found on 10.5281/zenodo.7555103.

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
