# Peer review of "The Mycobacterium bovis BCG GroEL1 Contributes to Isoniazid Tolerance in a Dormant-Like State Model"

_microorganisms, 2023, doi:10.3390/microorganisms11020286_

Round 1
Reviewer 1 Report
In my opinion the article is well written. Research methods are modern and correspond to the task. The conclusions are well founded.In principle, the article can be published as presented, but I have two small remarks:
1. Figure 2B. Divisions on the axis are practically not separated from each other. It would be nice to enlarge the picture.
2. supplementary. Figure S2. Please check the caption below the picture.
Author Response
Dear reviewer,
Thank you for your suggestions which are important to improve the presentation of our results. Indeed, we completed the Fig. legend of Suppl. Fig. 2 and the Fig. 2 was enlarged to allow better discrimination between the different conditions.
Reviewer 2 Report
The article "The Mycobacterium bovis BCG GroEL1 contributes to isoniazid tolerance in a dormant-like state model" describes how M. bovis BCG becames sensitive to isoniazid, when in a dormancy state, if lacks of the GroEL protein.
The article is based on previous authors studies and describes the different effect of Isoniazid on BCG during a common state of replication respect to the dormancy state, in both strains, with and without GroEL1.
The reported experiments are well described and the reasoning is clear and supported by evidences. Anyway, I personally feel the lack of identification of real mechanisms at the base of the sensitivity in dormancy state of the strain lacking of GroEL. Let me better explain my point: GroEL is a shaperonin, this means that it is involved in the folding or new synthetized proteins or in saving/mantaining the folding of already existing ones. The first thing that I thought reading the article is that probably, during persistance/dormancy it may be involved in mantaining the folding of some metabolic enzymes, probably enzymes of the ROS metabolism. In this sense it is only indirectly involved in the INH resistance at the reported condition. It would be interesting to read in future studies the identification of the direct resistance mechanism.
Have nothing much to say on how to improve the manuscript, just a couple of changing in text organization before pubblication. Following my advices:
line 118 - please check the paragraph, maybe there is an unwanted start of a new line.
line 287-288 - it's a bit confusing at a first view where the tagline ends and where the article text begins, you could little separate them.
Thanks for your work.
Author Response
Dear reviewer,
Thank you for your suggestions. Indeed, the spaces between paragraphs or between the text and the fig. legends were sometimes confusing. We improved it. Although GroEL1 might not be a classical chaperonin (no high oligomeric conformation and low ATPase activity), we can indeed not exclude this possibility of mechanism and this is now added in the last sentence of the Results and Discussion.